Exploring the differential stages of the pigment metabolism by pre-harvest bagging and post-harvest ethylene de-greening of Eureka lemon peel

Chen Anjun chen_anjun@sicau.edu.cn
Liu Lu
Liu Xiaoping
Fu Yunyun
Li Jie
Zhao Jianglin
Hou Xiaoyan
Sichuan Agricultural University , Yaan , China
Winkler Robert
Electronic publication date: 2021 Jun 8
Publication date: 2021
Volume: 9
Electronic Location ID: e11504
Received 2020 Apr 27; Accepted 2021 May 3
Copyright: ©2021 Anjun et al.
Copyright year: 2021
Copyright holder: Anjun et al.
License: This is an open access article distributed under the terms of the Creative Commons Attribution License, which permits unrestricted use, distribution, reproduction and adaptation in any medium and for any purpose provided that it is properly attributed. For attribution, the original author(s), title, publication source (PeerJ) and either DOI or URL of the article must be cited.
License URL: https://creativecommons.org/licenses/by/4.0/

Keywords: Chlorophyll degradation, Carotenoid biosynthetic, Genes, Individual carotenoid contents

Funding: National modern agricultural industry technology system Sichuan innovation team This work was supported by the National modern agricultural industry technology system Sichuan innovation team. The funders had no role in study design, data collection and analysis, decision to publish, or preparation of the manuscript.

==============================
Pre-harvest bagging or post-harvest ethylene treatments on lemons are commonly applied to change the surface color from green to favorable yellow. In this study, the differential mechanisms of the pigment metabolism by the two treatments were investigated by pigments contents and related genetic expression. The results showed that both treatments reduced the number of chloroplasts and the content of chlorophyll. The differential expression of PSY1 and PSY2 were observed, causing the different accumulation of the main carotenoid phytoene content. The differential expression of NYC resulted in altered contents of chlorophyll a and chlorophyll b, and further led to the difference in a* value. More interestingly, the degradation of chlorophyll uncovered the color of carotenoids, leading to the color changed from green to yellow.

Introduction

As one of the third most important Citrus species after orange and mandarin (Serna-Escolano et al., 2019), lemon (Citrus limon L. Burm.f.) is popular for its favorable flavor and natural ingredients like vitamins C (González-Molina et al., 2010) and phenolic compounds (Dong et al., 2019). Skin color is one of the key characteristics for citrus fruit, especially for lemons, because it largely affects the purchase intentions of consumers (Gao et al., 2019). Generally, customers prefer lemons with yellow color rather than green, however, lemons are usually harvested before fully ripe, when the fruit has the highest acid content, but the skin is still green (Zhang & Zhou, 2019). One practical operation to obtain the yellow color is bagging during the fruit growing process. Pre-harvest bagging is commonly used as a physical protection method to reduce the mechanical damage, fruit cracking, and make the fruit dye evenly (Yuan et al., 2019). It has also been widely used in other fruit such as grape, apple and kiwi (Guo et al., 2020; Yuan et al., 2019; Liao et al., 2019). Another way for the favorable yellow color is the ethylene treatment during the post-harvest process. Exogenous ethylene is regard as effective strategy to affect fruit color, and has been successfully applied to change the peel color from green to orange or yellow in navel orange and ‘Eureka’ lemon (Hu et al., 2018; Zhang & Zhou, 2019). Previous studies on fruit de-greening were mostly focused on optimization of fruit bags materials, ethylene concentration, de-greening temperature and time (Liao et al., 2019; Jomori et al., 2014). However, the differential expression of the pigment metabolism by the two de-greening methods has not been reported yet.

The color of lemon is mainly determined by two main classes of pigments: chlorophylls and carotenoids: the former one is mainly responsible for green color, while the latter is mainly for yellow (Rodrigo et al., 2013). The biosynthesis and degradation of chlorophylls, as can be seen in Fig. 1, includes four stages: (1) synthesis of Mg-protoporphyrin IX from glutamic acid through a series of reactions, and then convert to chlorophyll a; (2) interconversion of chlorophyll a and chlorophyll b, also known as chlorine leaf plant cycle; (3) chlorophyll bound; and (4) degradation of chlorophyll (Tanaka & Tanaka, 2006).

Figure 1 Pathway of chlorophyll degradation.

Genes styled in red are those studied in this work. Gene names are abbreviated as follows: HEMA, Glutamy 1-tRNA reductase; GAS, 2,1-aminotransferase; HEMB, d-Aminolevulinic acid dehydratase; HEMC, porphobilinogen deaminase; HEMD, Uroporphyrinogen III synthase; HEME, Uroporphyrinogen III decarboxylase; HEMF, Coproporphyrinogen III oxidase; CHLH, Mg-chelatase; NYC, chlorophyll b reductase; CAB, chlorophyll a/b binding protein; SGR, staygreen protein; Chlase, chlorophyllase; MCS, metal quelating substance; Pao, pheophorbide a oxygenase; RCCR, real chorophyll catabolite reductase; LHCP, chlorophyll bound.

Figure 2 shows the biosynthesis pathways of carotenoids. In the general isoprenoid biosynthesis pathway, isopentenyl pyrophosphate (IPP), which is derived from glucose molecules, is the first relatively direct precursor (Lu & Li, 2008). Though the isoprenoid pathway, IPP and dimethylallyl diphosphate (DMAPP) can be synthesized to geranylgeranyl diphosphate (GGPP), which is the central intermediate for carotenoids and other isoprenoid biosynthesis. Two molecules of GGPP can be condensed to phytoene (Li & Van, 2007), and subsequently, phytoene is dehydrogenated into ruthenium-carotene, and further dehydrogenated into lycopene (Isaacson et al., 2002). After that, there are two pathways downstream, affected by the catalysis of different enzymes. In the first pathway, α-carotene is synthesized by lycopene ε-cyclase (LCYe) and lycopene β-cyclase (LYCb1), and further generates lutein (Nisar et al., 2015), which is the most abundant carotenoid found in plant leaf tissue. As can be seen, LCYe plays a key role in determining the ratio of β-carotene to α-carotene (Harjes et al., 2008).

Figure 2 Pathway of carotenoid biosynthesis.

Red named genes are those studied in this work. Gene names are abbreviated as follows: PSY, phytoene synthase; PDS, phytoene desaturase; ZDS, ζ carotene desaturase; CRTISO, carotenoid isomerase; LCYe, lycopene ε-cyclase; LCYb1, lycopene □-cyclase 1; LCYb2, lycopene □-cyclase 2; CHYB, □-carotene hydroxylase; CCDs, carotenoid cleavage dioxygenase; ZEP, zeaxanthin synthase; VDE, violaxanthin de-epoxidase; NSY, neoxanthin synthase; NCEDs, 9-cis-epoxycarotenoid dioxygenase.

The biosynthesis and biodegradation of carotenoids and chlorophylls have been comprehensively investigated by classical genetic and molecular genetic analysis in some plants like pea and broccoli (Parveen et al., 2019; Luo et al., 2019). But there is still a lack of studies in lemon. Although pre-harvest bagging and post-harvest ethylene de-greening treatments can help change the lemon color from green to a favorable yellow, the gene expressions and metabolites of these two methods were likely to be different. The present study was therefore conducted to explore the differential mechanisms of the chlorophyll degradation and carotenoid biosynthesis in pre-harvest bagging and post-harvest ethylene de-greening of lemon peel.

Materials and Methods

Fruit materials

Eureka lemon (Citrus limon (L.) Burm.f.) trees were cultivated in a 12-year-old lemon orchard located at Anyue County, Sichuan Province, China. In August of 2018, young fruit was bagged with paper bags (Sichuan Meishan Guanrong Bag Making Co., Ltd.) (Wen, 2009). The Bagged and non-bagged fruit was grown under the same cultural practice. The fruit with similar quality (TSS >7.5) and weight between 140 g and 180 g was harvested at 2 November 2018. The farmer Shuyong Luo provided lemons and permission for fieldwork on privately owned land in Anyue County, Ziyang City.

Ethylene treatment and storage

Half of non-bagged green lemon fruit was treated by 5 µL L−1 ethylene at 28 °C, with a humidity of 95%, and less than 0.3% carbon dioxide for 5 d. The other half of non-bagged green lemons were used as control. Each treatment contained 3 biological replicates, and there were 20 fruit for each repetition. Color differences were compared with photos. The peels of 60 fruits in the three groups were collected, cut into small pieces, frozen by liquid nitrogen, and then stored at −80 °C until use.

Fruit surface color observation

Peel color was measured using HP-C226 Precision colorimeter (Shanghai, China) on four locations on the equator of each fruit.

Spectrophotometric determination of chlorophyll and total carotenoid content

Chlorophyll and total carotenoid were extracted according to the method reported by (Yuan et al., 2017). Briefly, 1.0 g peel was grounded to fine powders in liquid nitrogen. 10 mL acetone containing 0.01% BHT (2,6-di-tert-butylmethylphenol) was added, followed by centrifugation at 4, 500 ×g for 10 min at 4 °C. The supernatant was collected and the residue was extracted again by the same procedure until colorless. The supernatants were mixed and were adjusted to 25 mL with acetone containing 0.01% BHT. The pigments contents were calculated based on the absorbance at 470, 645 and 662 nm, respectively.

LC MS/MS analysis of carotenoids

After harvesting, the three group of lemon peels were frozen in liquid nitrogen, and stored at −80 °C until use. 100 mg sample was grounded into powder in liquid nitrogen, and extracted with a solution of n-hexane: acetone: ethanol at 2:1:1 (V/V/V). The extracts were vortexed for 30 s, an ultrasound-assisted extraction was carried out for 20 min at room temperature, and then the extracts centrifugated at 10, 000 ×g for 5 min. The supernatants were collected and the residue was extracted again by the same procedure. The supernatants were collected and evaporated to dryness under nitrogen gas flow. 200 µL 75% methanol was added to reconstitute the solution. The supernatant was collected for LC-MS analysis.

The LC MS/MS analysis of carotenoids was performed by Metware Biotechnology Co., Ltd. (Wuhan, China). A LC-ESI-MS/MS system (UHPLC, ExionLCTM AD; MS, Biosystems 6500 Triple Quadrupole) was used. The analytical conditions were as follows: HPLC: YMC C30 column (3 um, two mm*100 mm); solvent system: acetonitrile: methanol (3:1, V/V with 0.01% BHT, eluent A) and methyl tert-butyl ether with 0.01% BHT (eluent B); gradient program: 85:15 A:B at 0 min, 75:25 A:B at 2 min, 40:60 A:B at 2.5 min, 5:95 A:B at 3 min, 85:15 A:B at 6 min; flow rate: 0.8 mL min−1; temperature: 28 °C; injection volume: 5 uL. The effluent was alternatively connected to an ESI-triple quadrupole-linear ion trap (Q TRAP)-MS.

API6500QTRAP LC/MS/MS System, equipped with an APCI Turbo lon-Spray interface, was operated in a positive ion mode and controlled by Analyst 1.6.3 software (AB Sciex). The APCI source operation parameters were as follows: ion source: turbo spray; source temperature: 350 °C; curtain gas (CUR): 25.0 psi; the collision gas (CAD): medium. DP and CE for individual MRM transitions were done with further DP and CE optimization (Petry & Mercadante, 2018).

RNA extraction and RNA-Seq

RNA extraction and RNA-Seq were conducted by Pacino Biotech Co.,Ltd. (Shanghai, China). The original off-machine data was filtered by removing the reads with a length of less than 50 bp and a sequence average quality below Q20. The obtained high-quality sequence was spun-headed to a transcript sequence. The transcripts were clustered, the longest transcript was selected as Unigene. Unigene of each sample was obtained to investigate the expression difference analysis, enrichment analysis, cSNP and InDel analysis, including GO, KEGG, eggNOG, SwissProt, Pfam annotation and ORF prediction, SSR prediction, etc.

Real-time quantitative PCR(RT-qPCR) analysis

The primers for corresponding sequences were designed according to literatures and were listed in Table 1. CitActin was used as the reference gene. qRT-PCR was performed using TIB8600 (Taipu Biosciences Co., Ltd.). Each reaction contained 10 µL 2 ×SYBR real-time PCR premixture, 0.4 µL PCR forward primer (10 µM), 0.4 µL PCR reverse primer (10 µM), 1µL cDNAs, and RNase free ddH2O to a final volume of 20 µL. The PCR conditions were 95 °C for 15 min, followed by 40 cycles of 95 °C for 15 s and 60 °C for 30 s. Three biological replicates were performed, and the mean values were used for RT-qPCR analysis. The relative expression of the genes was calculated according to the method of 2−ΔΔ Ct.

Table 1 PCR primers used in this study.

Genes	Forward	Reverse	
CitActin	CCAAGCAGCATGAAGATCAA	ATCTGCTGGAAGGTGCTGAG	
Chlorophyll metabolism related genes			
CitCHLH	CGATGTTCGTGAAGCAGCAACTC	TTGGAATGTGGCGTCTGCTGTGC	
CitNYC	GGCACGGTTTTCCTTTACAGATG	TGTTGTAGTTCTGACGCTTTCTG	
CitChlase	GTGGGATTGTGGTGGCGTTTCT	ACTTTTACATGAGTTGTCGTAAGC	
CitPao	CAGCACACCCTCAAGTGTTCATC	AAACAAAGGGAATACTGAGGAAAC	
CitSGR	CAACTGTTGCTTTCCTCCAATGAG	TAAAACCCCACCAATACTTTG	
CitCAB1	ATCCATTGGGTTTGGCTGATGAC	AACTCTTATCAACCGAAGCTCACT	
CitCAB2	CCGTCTGGCTATGTTCTCCATGT	AGATGAAACAGACACCATCAAGTC	
CitRCCR	AGTGTGCTTGTGGAGGGAGACAT	AATTAAGGTCAATCAACTCCAATTC	
Carotenoid metabolism related genes			
CitPSY1	CGTTGATGGGCCTAATGCTT	ACCTGGACTCCCACCTGTCTAA	
CitPSY2	GATTCCTCAGCTTCTGCTC	CTTGCTCTTGTAATTTGCTCT	
CitPDS	TGGCAACCCCCCAGAGA	CACCCAGTGACTGAATGTGTT	
CitZDS	AAAGGCACTTGTTGATCCTGATG	ACCAATCAGAGAAGCTTATACTATCCA	
CitCRISO	AAAGACACACCGGCGGTATC	CGAGGCATTGGCCCATAG	
CitLCYb2	CCTTGGCTCAACCAGGATGA	ACCCATTCCACACTTTCTGATGA	
CitLCYe	AAGGTGTGTCGAGTCAGGTGTTT	CCACTGGTAGATTCCGTAATGCT	
CitCHYB	GCGGCTCACCAGCTTCAC	CCGAGAAAGAGCCCATATGG	
CitZEP	CTAAAGAGCTATGAGAGAGCTAGGAGACT	CACTGCGGCCGATCTTG	
CitVDE	CAAAGACTTCAATGGGAAGTGGTA	TGGCAATCAAAAGTATCGAAGGA	
CitCCD1	TTCATGGTCTGCGCATCAA	GACGTGAAGTCCTCACAAAACG	
CitNCED1	AACCCGTCTGCCAGAACCTT	GTTGGCTCCGTTTCTGACGTA	
CitNCED2	GGTGCCAACCCATTATTCGA	GCCGTCACCGTCAAAGAAAT	
CitNCED3	GCTTCCGTTTGTGGCCTACTT	ATTGACCCGGCATTTTTATGTG	

Statistical analysis

Three biological replicates (n = 3 × 3) were performed for all treatments. The data were analyzed with SPSS 16.0 and Excel 2013, and the results were expressed as mean values with standard deviation (SD). The difference was considered to be significant at P < 0.05.

Results

Fruit surface color changes

The effects of pre-harvest bagging and post-harvest ethylene treatment on lemon surface color were shown in Fig. 3. It can be seen that, after the treatments, the color of lemon peel was changed from green to yellow. The L*, a*, b* color parameters (Fig. 4) was corresponded to the visual observation. The a* values of ethylene and bagged fruit were significantly higher than that of control. Control and ethylene treated fruit showed lower L* value than bagging group. There were no significant changes in b* value for the three groups.

Figure 3 The picture of the lemon.

Figure 4 Effect of pre-harvest bagging and post-harvest ethylene de-greening treatments on chromatic aberration.

L* the lightness ranging from black to white; a* a scale ranging from green to red; and b* a scale ranging from blue to yellow. Vertical bars represent standard deviations of the three independent biological replicates means. Different letters are statistically different by the Duncan’s multiple range (P < 0.05).

Chlorophyll and carotenoid content

As shown in Fig. 5, the chlorophyll a and chlorophyll b content in bagging and ethylene fruit were significantly lower than that in control (P < 0.05), while for the carotenoid content, ethylene treated fruit were significantly higher than bagging group, and control was the lowest one (P < 0.05). These results indicated that the ethylene and bagging treatment could degrade the chlorophyll a and chlorophyll b content, and improve the carotenoid synthesis.

Figure 5 Effect of pre-harvest bagging and post-harvest ethylene de-greening treatments on pigment.

Vertical bars represent standard deviations of the three independent biological replicates means. Different letters are statistically different by the Duncan’s multiple range (P < 0.05).

Carotenoid composition

Table 2 showed the carotenoid composition by different treatments. The results showed that bagging and ethylene exhibited different effects on the accumulation of individual carotenoids. Bagging and ethylene treatment sharply increased the content of phytoene and capsanthin (P < 0.05), but decreased the content of lutein, α-carotene, β-carotene and other substances (P < 0.05). The results indicated that the yellow color of Eureka lemon might be mainly attributed by phytoene. The content of γ-carotene, phytoene and capsanthin in the ethylene group were obviously lower than that in the bagging group, whereas the content of lutein, neoxanthin, zeaxanthin, violaxanthin and antheraxanthin and capsorubin were the opposite.

Table 2 Effect of pre-harvest bagging and post-harvest ethylene de-greening treatments on individual carotenoid contents.

	Groups	
Carotenoids	Control (µg g−1)	Bagging (µg g−1)	Ethylene (µg g−1)	
α-carotene	0.33 ± 0.019a	0.06 ± 0.001b	0.06 ± 0.003b	
α-cryptoxanthin	0.22 ± 0.018a	0.17 ± 0.000b	0.17 ± 0.000b	
β-carotene	3.45 ± 0.234a	0.09 ± 0.014b	0.24 ± 0.016b	
β-apoline aldehyde	0.11 ± 0.004a	0.06 ± 0.003b	0.06 ± 0.005bc	
β-cryptoxanthin	3.10 ± 0.26a	0.07 ± 0.041b	0.27 ± 0.078b	
γ-carotene	0.34 ± 0.019a	0.17 ± 0.007b	0.15 ± 0.006c	
ε-carotene	ND	ND	ND	
phytoene	3.71 ± 0.433c	120.53 ± 9.305a	14.17 ± 2.016b	
Hexahydro lycopene	ND	ND	ND	
lutein	15.62 ± 0.785a	0.10 ± 0.022c	1.73 ± 0.148b	
neoxanthin	0.26 ± 0.006a	ND	0.03 ± 0.005b	
zeaxanthin	2.55 ± 0.231a	1.36 ± 0.169c	2.00 ± 0.238b	
lycopene	0.65 ± 0.023a	0.69 ± 0.051a	0.64 ± 0.039a	
violaxanthin	2.07 ± 0.339a	0.42 ± 0.050c	0.76 ± 0.050b	
antheraxanthin	0.27 ± 0.021a	0.01 ± 0.001c	0.065 ± 0.006b	
astaxanthin	ND	ND	ND	
capsanthin	ND	0.12 ± 0.009a	0.03 ± 0.000b	
capsorubin	0.02 ± 0.000b	0.02 ± 0.000c	0.03 ± 0.001a	
Notes.

All values were the mean  ± SD of three independent biological replicates measurements. The different small letters in the same row indicated significant differences at the 0.05 level, and ND represented not detected.

Screening carotenoid biosynthesis and chlorophyll degradation related genes in the transcriptome

The genes involved in chlorophyll degradation and carotenoid biosynthesis were analyzed. 22 genes that encoded enzymes in the carotenoid biosynthetic and chlorophyll metabolic pathway were screened in lemon transcriptome: CHLH, NYC, Chlase, Pao, SGR, CAB1, CAB2, RCCR, PSY1, PSY2, PDS, ZDS, CRTISO, LCYb1, LCYb2, CHYB, ZEP, VDE, CCD1, NCED1, NCED2, NCED3, LCYe. The heat map of the carotenoid biosynthetic and chlorophyll metabolic related genes was drawn according to FPKM in Fig. 6. It can be seen that control and ethylene groups were more closely related than bagging group. PSY1, RCCR, CHLH, CCD1, SGR, NCED1, Chlase, ZDS, Pao, LCYE and CAB2 were closely related in a cluster, while the other genes were clustered in another group.

Figure 6 Heat map diagram of expression levels for chlorophyll and carotenoid degradation pathway genes analyzed by KEGG.

T means bagging, Q means control, Z means ethylene.

In this transcriptome, two genes (PSY1 and PSY2) in ethylene group recorded a more expression than that in control and bagging groups. Furthermore, nine genes (CHYB, NCED3, NCED2, CRTISO, PDS, LCYB2, VDE, NYC) in bagging group expressed more than that in control and ethylene groups. And eleven genes (RCCR, CHLH, CCD1, SGR, NCED1, Chlase, ZDS, Pao, LCYE, CAB2, CAB1) in control group expressed more than that in ethylene and bagging group. Except for NYC, the expression of the chlorophyll metabolism pathway genes (CHLH, Chlase, Pao, SGR, CAB1, CAB2, RCCR) were down-regulated after the peel turned yellow.

RT-qPCR analysis of transcription levels of chlorophyll and carotenoid degradation pathway genes

As shown in Table 3, the genetic expressions of 22 key carotenoid biosynthetic and chlorophyll degradation pathway genes in different groups were studied using RT-qPCR analysis. The expression of CHLH, Chlase, Pao, CAB1, CAB2 and RCCR in bagging and ethylene groups were lower than that in control group. All of the expression of the chlorophyll degradation related genes, except RCCR, in bagging group were higher than that in ethylene group. NYC was up-regulated after pre-harvest bagging treatment. PSY gene is a multi-gene family, the present paper studied two members of CitPSY1 and CitPSY2. In the carotenoid biosynthetic pathway, the expression of PSY1, PDS, ZDS, CRTISO, LCYb2, CHYB, ZEP, VDE, CCD1, NCED1, NCED2, NCED3 and LCYe in ethylene group were lower than that in control group. The expression of all genes, except PSY2, in ethylene group were lower than that in bagging group. The results also showed that PSY1, CRTISO, VDE, NCED1, NCED2 and NCED3 were up-regulated after bagging treatment.

Table 3 Effect of pre-harvest bagging and post-harvest ethylene de-greening treatments on genes expression associated with carotenoid and chlorophyll metabolism.

	Control	Bagging	Ethylene	
CitCHLH	1.31 ± 0.13a	1.03 ± 0.12b	0.56 ± 0.08c	
CitNYC	0.43 ± 0.05b	1.08 ± 0.16a	0.19 ± 0.02c	
CitChlase	3.39 ± 0.29a	0.82 ± 0.18b	0.40 ± 0.05c	
CitPao	1.58 ± 0.20a	1.13 ± 0.15b	0.90 ± 0.09c	
CitSGR	0.75 ± 0.12b	1.08 ± 0.20a	0.59 ± 0.09c	
CitCAB1	2.86 ± 0.45a	1.07 ± 0.11b	0.46 ± 0.06c	
CitCAB2	3.51 ± 0.44a	1.04 ± 0.11b	0.91 ± 0.94c	
CitRCCR	2.46 ± 0.30a	1.27 ± 0.44c	1.90 ± 0.31b	
CitPSY1	0.70 ± 0.10b	1.05 ± 0.07a	0.57 ± 0.08c	
CitPSY2	3.84 ± 0.40a	0.76 ± 0.19c	2.74 ± 0.39b	
CitPDS	1.61 ± 0.17a	1.03 ± 0.12b	0.81 ± 0.08c	
CitZDS	2.48 ± 0.46a	1.11 ± 0.16b	1.07 ± 0.14b	
CitCRTISO	1.00 ± 0.19a	1.12 ± 0.27a	0.78 ± 0.12b	
CitLCYb2	2.66 ± 0.27a	1.40 ± 0.45b	0.73 ± 0.13c	
CitCHYB	1.60 ± 0.09a	1.44 ± 0.37a	0.73 ± 0.18b	
CitZEP	3.02 ± 0.65a	1.63 ± 0.59b	1.05 ± 0.33c	
CitVDE	0.63 ± 0.104b	0.82 ± 0.15a	0.51 ± 0.09c	
CitCCD1	0.97 ± 0.17a	0.93 ± 0.13ab	0.77 ± 0.18b	
CitNCED1	0.69 ± 0.13b	0.85 ± 0.18a	0.27 ± 0.08c	
CitNCED2	1.01 ± 0.14a	1.08 ± 0.15a	0.23 ± 0.04b	
CitNCED3	0.54 ± 0.11b	1.53 ± 0.72a	0.12 ± 0.01c	
CitLCYe	1.71 ± 0.27a	1.04 ± 0.08b	0.89 ± 0.12b	
Notes.

All values are the mean ± SD of at least three independent biological replicates measurements. The different small letters in the same row indicate significant differences at the 0.05 level, and ND indicates not detected.

Discussion

The surface color of Citrus fruit is one of the most important quality characteristics that is a decisive factor for consumer acceptance (Rodrogo et al., 2013). Pre-harvest bagging treatment and post-harvest ethylene de-greening treatment can help change the lemon color from green to favorable yellow. Similarly as many other fruit, like kiwi and ‘Valencia Delta’ orange (Zhang et al., 2018; Pereira, Machado & Costa, 2016), the color of lemon peel is highly related to the changes of chlorophyll and carotenoid content (Yuan et al., 2017). The present study showed that chlorophyll a and chlorophyll b contents were declined after the yellowing process. The content of chlorophyll a in bagging group was higher than that in ethylene group, whereas the content of chlorophyll b was just the opposite. Although the bagging and ethylene fruits still contained ca. 50% of the Chlorophyll of the control treatment. The ratio of chlorophyll to carotenoids was 9.39 in the control group, 2.79 in the bagging group, and 2.03 in the ethylene group. The reason is not clear and need further study.

NYC protein has chlorophyll b reductase activity in vitro, and regulate the thylakoid membrane degradation (Cheng et al., 2012). In this study, NYC was up-regulated after pre-harvest bagging treatment, leading to higher degradation of chlorophyll b to chlorophyll a. This phenomenon explained the reason why the bagging group recorded higher chlorophyll a content but lower chlorophyll b content than ethylene group. Except for NYC, the expression of the chlorophyll metabolism pathway genes (CHLH, Chlase, Pao, SGR, CAB1, CAB2, RCCR) were down-regulated after the peel turned yellow. This result was accord to the other study in mandarin fruit (Yuan et al., 2017).

Bagging and ethylene treated lemons showed yellower color than control, and recorded higher content of phytoene and capsanthin, but lower content of lutein, α-carotene, β-carotene and other substances than control group, suggesting that the yellow color of Eureka lemon may be mainly derived from phytoene. This could be primarily explained by the high expression of PSY and the low expression of the desaturase PDS (Kato et al., 2004). In the “green” lemon peel, the colored carotenoids were main lutein, which little influenced the yellow color of the lemon peel. Although this pigment was found in high quantities, the degradation pattern was similar as chlorophylls. When the peel was green, the concentration of lutein reached values up to 15.62 µg g−1, but its color was masked by the chlorophylls. However, when they degraded, a decrease in lutein was also found, indicating that this pigment barely contributed to the color of lemon peel (Conesa et al., 2019). During the coloration process, the rapid loss of the chlorophyll played a leading role for the shift of fruit color from green to yellow, despite the content of the carotenoids (lutein, α-carotene, β-carotene, et al.) decreased in the lemon peel (Shen et al., 2019).

The synthesis of carotenoid in the bagging and ethylene groups was analyzed based on the genetic expression and the changes of carotenoids content. The carotenoid levels were shown to be dependent on the PSY expression, which was the first rate-limiting enzyme in the carotenoid biosynthetic pathway presented in many plants such as tomato, apple, carrot, papaya and pepper (Fu et al., 2018; Heng et al., 2019; Fantini et al., 2013; Ampomah-Dwamena et al., 2015; Fuentes et al., 2012; Shen et al., 2019; Wei et al., 2019), it plays an important role in controlling the metabolism of the carotenoids. PSY1 is chromoplast-specific and its expression is extremely high during ripening stage of fruit, while PSY2 acts predominantly in chloroplast-containing tissues and does not have a contribution to the production of carotenoid in fruit (Cao et al., 2019). In our study, PSY1 was up-regulated after bagging treatment, and the expression of PSY2 in ethylene group was higher than that in bagging group. This phenomenon could explain why the phytoene content in bagging group was higher than that in ethylene group. Violaxanthin de-epoxidase (VDE) and zeaxanthin epoxidaze (ZEP) are the two enzymes in the xanthophyll cycle. In excess light conditions, VDE catalyzes the conversion of violaxanthin to zeaxanthin via antheraxanthin, whereas ZEP catalyzes the reverse reaction (Shen et al., 2019). LCYE plays a decisive role in the ratio of α-carotene and β-carotene. The present study showed that high LCYE expression resulted in high luteincontent, which was consistent with the study in marigold (Cheng et al., 2019). The expression of the LCYE was lowered in both the bagging and ethylene groups. These results revealed that even though both pre-harvest bagging and post-harvest ethylene treatment could help change the lemon color from green to yellow, the mechanisms of the two treatments were largely different.

Conclusions

This work mainly compared the differential mechanisms of chlorophyll degradation and carotenoid biosynthesis by pre-harvest bagging and post-harvest ethylene de-greening treatments, The results showed that both de-greening treatments can help to change the lemon peel color from green to favorable yellow. The expression of key genes and pigments contents in the lemon peel during the yellowing process revealed that carotenoid biosynthetic and chlorophyll degradation pathway by the two treatments were different. Our results clearly showed that the differential expression of PSY1 and PSY2 caused a difference in the main carotenoid phytoene by two treatments. The differential expression of NYC resulted in different contents of chlorophyll a and chlorophyll b, which showed a difference in a* value. Like other yellow citrus, the degradation of chlorophyll, which made the color of carotenoids appear, was the main reason for the changes of the lemon peel color from green to yellow.

Supplemental Information

Supplemental Information 1 Raw data

Click here for additional data file.

Additional Information and Declarations

Competing Interests

Author Contributions

Field Study Permissions

Data Availability

The authors declare there are no competing interests.

Chen Anjun and Xiaoyan Hou conceived and designed the experiments, authored or reviewed drafts of the paper, and approved the final draft.

Lu Liu conceived and designed the experiments, performed the experiments, analyzed the data, prepared figures and/or tables, and approved the final draft.

Xiaoping Liu, Fu Yunyun, Jie Li and Jianglin Zhao performed the experiments, prepared figures and/or tables, and approved the final draft.

The following information was supplied relating to field study approvals (i.e., approving body and any reference numbers):

The farmer Shuyong Luo provided lemons and permission for fieldwork on privately owned land in Anyue County, Ziyang City.

The following information was supplied regarding data availability:

The RNA sequencing data are available at the Sequence Read Archive (SRA) at NCBI: BioProject ID PRJNA606613.

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
