# Peer review of "Exploring the differential stages of the pigment metabolism by pre-harvest bagging and post-harvest ethylene de-greening of Eureka lemon peel"

_PeerJ, doi:10.7717/peerj.11504_

## Round 0.1 · original submission · Major Revisions

Both reviewers suggest a thorough revision of previous reports (and your preferences. As well, you need to resolve various comments on your methods and data. In addition, some additional data should be presented on the molecular basis of the 'de-greening' mechanism (carotenoid profiles, Chl a/Chl b ratio, microscopy).

Reviewer 1 ·

Basic reporting

The present manuscript provides some new and interesting information and could be suitable for publication in PeerJ after considering the following comments.
In general, the experiment was well performed and results are clear and according to conclusions.

Experimental design

See comments below.

Validity of the findings

See comments below.

Additional comments

Dear Editor, in the manuscript PeerJ-48058 authors evaluated the chlorophyll and individual carotenoids contents as well as the expression of the genes involved in their biosynthesis and degradation in lemon fruit after bagging during on-tree development or treated with ethylene after harvest. Results show that both treatments were effective on inducing lemon colour changes from green to yellow, due to reduction in chlorophyll and increases in carotenoid concentration, although differences exist on the expression of some genes involved in these metabolic pathways.
The use of ethylene treatment for degreening lemon fruits has been reported in several previous papers, which should be considered in this manuscript, such as the following ones, among others:
- Mayuoni, L., Porat, R. 2011. Postharvest treatments for degreening of 'Villa franca' lemons. HortTechnology, 21(5), pp. 624-627.
- Zhang, P., Zhou, Z. 2019. Postharvest ethephon degreening improves fruit color, flavor quality and increases antioxidant capacity in ‘Eureka’ lemon (Citrus limon (L.) Burm. f.). Scientia Horticulturae, 248, pp. 70-80.
- Porras, I., Brotons, J.M., Conesa, A., (...), Pérez-Tornero, O., Manera, F.J. 2015. Quality and fruit colour change in Verna lemon. Journal of Applied Botany and Food Quality, 88, pp. 215-221.
- Conesa, A., Brotons, J.M., Manera, F.J., Porras, I. 2014. The degreening of lemon and grapefruit in ethylene atmosphere: A cost analysis. Scientia Horticulturae, 179, pp. 140-145.
However, literature about the use of bagging to accelerate colour changes in lemon fruits is scarce, although there is information about other citrus fruit, such as oranges, pomelo, lime, etc. Then, the present manuscript provides some new and interesting information and could be suitable for publication in PeerJ after considering the following comments:
On the other hand, information about carotenoid profile in lemon flavedo and changes during maturation is also scarce, although the following paper should be considered in discussion section:
- Multari, S., Licciardello, C., Caruso, M., Martens, S. 2020. Monitoring the changes in phenolic compounds and carotenoids occurring during fruit development in the tissues of four citrus fruits. Food Research International, 134,109228.
- Line 20: Use italic font for scientific names.
- Line 26: Add reference to support this statement.
- Line 28: Provide reference.
- Line 43: This reference appears as Tanaka and Tanaka in reference ist. Check and correct properly.
- Line 46: Lu and Li, according to reference list.
- Line 49: Li and Van, according to reference list.
- Line 53: Abbreviations for these enzymes are LCYe and LYCb1 in Figure 2. Be consistent, and use the same abbreviation for each one along the manuscript.
- Line 72: What quality parameters were used?
- Line 81: Consider adding “of the 20 fruits for each replicate” after each.
- Line 122: Provide reference for the procedure used to quantify individual carotenoids if available.
- Line 147: n= 3x3 or n=3’?P
- Line 160: It should be better to describe just L*, a*, b* colour parameters instead of chromatic aberration index.
- Lines 168-169: Check this sentence, because the contrary os observe in Fig. 5,.
- Lines 197-198, 205, 220-222: Move to discussion section.
- Line 221: This reference is missing in reference list. Should it be 2018?
- Line 231: Provide reference for colour and pigment in lemon flavedo.
- Line 234: These were not significant, according to Fig. 4.
- Line 238: This reference is missing in reference list.
- Line 239: Consider re-writing as “leading to higher degradation of chlorophyll b to chlorophyll a”
- Line 269: If ZEP was higher in bagging than in ethylene group, higher content of violaxanthin would be expected in bagging group than in ethylene group and the contrary is shown in Table 2.
- Line 288: An increase was also found in carotenoid content.
- Line 360: Move this reference to its correct place, according to alphabetical order.
- Line 370: Add “Serrano, M.” before 2019.
- Head of table 3: Delete “at least” because 3 replicates were used according to line 141.

·

Basic reporting

No comments.

Experimental design

See in 'comments to authors'

Validity of the findings

a) L. 157-159 – disappearance of chloroplasts; From what we know, chloroplasts do not disappear, they turn into chromoplasts (Rodrigo et al 2013), light microscopy should be replaced by EM.
b) L. 286-288. This is not surprising. It has been shown in early citrus carotenoid studies,
that in yellow citrus cultivars ( lemon, white grapefruit),where there is no prominent accumulation of darker pigments, the degradation of Chl exposes the yellow color of the peel. Apparently are not familiar with classic citrus lliteratue
c) L. 234- 236. Why not calculate the Chl a/Chl b ratio to prove this point?

Additional comments

a) More information on the 'bagging' treatment must be provided. Any previous report of this technique with citrus? What is the light and gas environment in the bag? Any ethylene in the bag?

b) A more basic comment: In my opinion, the 3 treatments represent different stages in the course of chloroplast-chromoplast transformation; the green control at the earliest stage (high Chlase and Pao activity), bagging and ethylene treatments - already degreened (low Chlase and Pao). This leads to a different interpretation of the gene expression data (Fig. 6). Thus, the claim of 'different metabolic pathways' has not been convincingly demonstrated, in my opinion, at least for the chlorophylls.
c) There are a lot of mistakes in the References.
d) A very thorough revision, perhaps with some experimental work will be required to enable acceptance of this MS for publication.

---

## Round 0.2 · Minor Revisions

Both reviewers recommend minor changes to your manuscript. Several references and figures 3 and 5 require corrections.

Reviewer 1 ·

Basic reporting

In the manuscript PeerJ-48058-R1 authors studied the differential mechanisms of the pigment metabolism on pre-harvest bagged or post-harvest ethylene treated lemons by measuring individual carotenoid content and related genetic expression. In general, the experiments were well performed and the manuscript is clearly written in professional and unambiguous language showing new and interesting information.
The manuscript could be suitable for publication although the following comments should be considered:

Experimental design

- Line 26: It should be Zhang and Zhou instead of Zhang et al. Check this cite along the manuscript.
- Line 59: Luo et al., 2019 is not found in reference list.
- Line 72: Please, clarify if lemon fruits were treated with ethephon or with ethylene. According to line 78, fruits were treated with ethylene at 5 mg L-1. Ethylene is a gas compounds so that concentration applied should be expresses in volume units.
- Line 83: Were samples from control and ethylene treated fruits taken after the 5 d of ethylene treatment?
- Line 103: Ethylene treated fruits were also used for carotenoids quantification. Please, clarity.

Validity of the findings

- Lines 159-160: This information should be moved to discussion section.
- Line 299: Move this reference after Cao et al., 2019.
- Line 328: Move this reference after Heng et al.
- Line 363: Add journal volume and pages.
- Check reference list and write references according to the journal format and use italic forn for scientific plant names.

Additional comments

See comments above.

·

Basic reporting

No comment

Experimental design

No comments.

Validity of the findings

No comment.

Additional comments

Although the principal findings of this report are O.K. , in my opinion the MS can not be published without further revision.
Here are my reservations:
a) Fig. 3. The light micrographs are of an inferior quality and cannot be published,
as said in my 1st review.
b) Fig 3 has no legend ? When was the photo taken? It seems that it shows fruit with almost no Chlorophyll, but in Fig. 5 the 'ethylene' and 'bagging' treatments still have 50% or more of the Chl. in the 'control'.
c) Fig, 5. It seems that the Chl a/b ratio of these 'control' fruits was lower than 1.
This is strange; any previous records of this phenomenon.
d) Assuming that Fig. 5 represents the time point that fruit samples were taken, the fruit were still in the midst of Chl. degradation, so there is no reason to think that the Chl catabolism genes have already finished their role( as stated in lines 262-263). Also not clear why NYC should behave differently than other catabolic genes.
e) Has any attempt been made to evaluate the light and, in particular, the atmosphere in bagged lemons, any ethylene? data from other fruits are not enough.
In conclusion, while the difference between 'bagging' and 'ethylene' in the Chl. and Carotenoids metabolism has been demonstrated a lot of things in the Results and in the Discussion need to be corrected.

---

## Round 0.3 · Minor Revisions

Dear authors, please resolve the comments about the color change and the respective Clorophyll content of the fruits.

Reviewer 1 ·

Basic reporting

Please, see comments below.

Experimental design

Please, see comments below.

Validity of the findings

Please, see comments below.

Additional comments

The manuscript has been revised and modified according to the reviewers' suggestions and it could be suitable for publication in its present form.

·

Basic reporting

OK

Experimental design

OK

Validity of the findings

OK

Additional comments

Re Reviewing MS 48058
Dear Authors / Editor
I have reread your MS. There is one comment I made in my previous review and remains unsettled.
In Fig 3 the BAGGING and ETHYLENE fruits are completely yellow, with no visible Chlorophyll. How can this be reconciled with the data of Fig. 5, where the BAGGING and ETHYLENE fruits still contained ca. 50% of the Chlorophyll of the CONTROL treatment?
This contradiction must be resolved prior to publication.
It seems to me that, unlike stated in the MS, the photos of Fig. 3 were taken at a later date, when the BAGGING and ETHYLENE fruits finished their color change.
I apologize for not using your Peer J review form.
Sincerely,
Eliezer E. Goldschmidt

---

## Round 0.4 · accepted · Accept

Thanks for your improved version.

Please revise the grammar of the new text in the proof of your manuscript.

'Although the bagging and ethylene fruits still contained ca. 50% of the Chlorophyll of the control treatment. The ratio of chlorophyll to carotenoids was 9.39 in the control group, 2.79 in the bagging group, and 2.03 in the ethylene group. The reason is not clear and need further study.'

Suggestions:
- Remove 'Although the'
- needs further studies